# On Frank-Wolfe Adversarial Training

**Theodoros Tsiligkaridis** [* 1]   **Jay Roberts** [* 1]

## Abstract

We develop a theoretical framework for adversarial training (AT) with FW optimization (FW-AT) that reveals a geometric connection between the loss landscape and the distortion of $\ell_\infty$ FW attacks (the attack's $\ell_2$ norm). Specifically, we show that high distortion of FW attacks is equivalent to low variation along the attack path. It is then experimentally demonstrated on various deep neural network architectures that $\ell_\infty$ attacks against robust models achieve near maximal $\ell_2$ distortion. To demonstrate the utility of our theoretical framework we develop FW-AT-Adapt, a novel adversarial training algorithm which uses simple distortion measure to adapt the number of attack steps during training. FW-AT-Adapt provides strong robustness against white- and black-box attacks at lower training times than PGD-AT.

## 1. Introduction

It is well known that small, carefully chosen input perturbations, known as adversarial perturbations, can fool deep neural networks (DNNs) into making incorrect predictions (Goodfellow et al., 2015). Various methods have been proposed to defend against such perturbations (Kurakin et al., 2017; Ros & Doshi-Velez, 2018; Madry et al., 2018; Moosavi-Dezfooli et al., 2019). One of the best performing algorithms is adversarial training (AT) (Madry et al., 2018), which is formulated as a robust optimization problem (Shaham et al., 2018). Computation of optimal adversarial perturbations is NP-hard (Weng et al., 2018) and approximate methods are used to solve the inner maximization. The most popular approximate method is projected gradient descent (PGD). Frank-Wolfe (FW) optimization has been recently proposed in (Chen et al., 2020) and was shown to effectively fool standard networks with less distortion. Recent work has shown that FW optimization can be effi-

*Equal contribution [1]MIT Lincoln Laboratory, Lexington, MA USA. Correspondence to: T. Tsiligkaridis <ttsili@ll.mit.edu>.

*Accepted by the ICML 2021 workshop on A Blessing in Disguise: The Prospects and Perils of Adversarial Machine Learning.* Copyright 2021 by the author(s).

ciently used to generate sparse counterfactual perturbations to explain model predictions and visualize principal class features (Roberts & Tsiligkaridis, 2021).

In this paper, we use the Frank-Wolfe optimization to derive a relationship between the $\ell_2$ norm of $\ell_\infty$ adversarial perturbations (distortion) and the structure of the loss landscape; based on this relationship we propose an adaptive Frank-Wolfe adversarial training (FW-AT-Adapt) method to obtain robustness. This method can achieve robustness near or above multistep PGD-AT while significantly decreasing training time.

## 2. Background and Previous Work

One of the most popular and effective defenses against adversarial attacks is adversarial training (AT) (Madry et al., 2018) which minimizes the adversarial risk

$$\min_\theta \mathbb{E}_{(x,y)\sim\mathcal{D}} \left[ \max_{\delta \in B_p(\epsilon)} \mathcal{L}(x + \delta, y; \theta) \right]. \qquad (1)$$

This framework was extended in the TRADES algorithm (Zhang et al., 2019) which proposes a modified loss function that captures the clean and adversarial accuracy tradeoff. Local Linearity Regularization (LLR) (Qin et al., 2019) uses an analogous approach where the adversary is chosen to maximally violate local linearity based on a first order approximation to (1).

To construct adversarial attacks at a given input $x$, AT uses Projected Gradient Descent (PGD) to approximate the inner maximization using a fixed number of iterations:

$$\delta_{k+1} = P_{B_p(\epsilon)}\left(\delta_k + \alpha\nabla_\delta\mathcal{L}(x + \delta_k, y; \theta)\right) \qquad (2)$$

where $P_{B_p(\epsilon)}$ is the orthogonal projection onto the constraint set. We refer to AT using $K$ step PGD as PGD(K)-AT. The computational cost of this method is dominated by the number of steps used to approximate the inner maximization. Using fewer PGD steps lowers this cost, but these amount to weaker attacks possibly leading to gradient obfuscation (Papernot et al., 2017; Uesato et al., 2018), a phenomenon where networks learn to defend against gradient-based attacks by making the loss landscape highly non-linear, and less robust models.

Other works have modified the number of steps used to approximate (1) such as curriculum learning (Cai et al., 2018) which monitors adversarial performance during training and increases the number of attack steps as performance improves, and (Wang et al., 2019) which use a FW convergence criterion to adapt the number of attack steps at given inputs. Both of these methods use PGD to generate adversarial examples and do not report improved training times.

We present Frank Wolfe Adversarial Training (FW-AT) which replaces the PGD inner optimization with a Frank-Wolfe optimizer. FW-AT achieves similar robustness as its PGD counterpart. Using a closed form expression for the FW attack path we derive bounds on the gradient alignment of the loss and attack path variation in terms of the distortion of the attack. This key insight leads to a simple modification of FW-AT where the step size at each epoch is adapted based on the $\ell_2$ distortion of the attacks and is shown to provide strong robustness and faster training times.

### 2.1. Frank-Wolfe Adversarial Attack

The Frank-Wolfe (FW) optimization algorithm has its origins in convex optimization though recently has been shown to perform well in more general settings (Frank & Wolfe, 1956; Jaggi, 2013). FW first optimizes a linear approximation to the original problem, called a Linear Maximization Oracle (LMO), $\bar{\delta}_k = \text{argmax}_{\delta \in B_p(\epsilon)} \langle \delta, \nabla_\delta \mathcal{L}(x + \delta_k, y) \rangle$. After calling the LMO, FW takes a step using a convex combination with the current iterate, $\delta_{k+1} = \delta_k + \gamma_k(\bar{\delta}_k - \delta_k)$ where $\gamma_k \in [0, 1]$ is the step size. An effective choice is $\gamma_k = c/(c + k)$ for some $c \geq 1$. In our experiments robustness of FW-AT was not sensitive to choice of $c$. This is in contrast to PGD-AT which can be highly sensitize to choice of step size (Wong et al., 2020). The LMO can be computed exactly for any $\ell_p$ and for the $\ell_\infty$ case, LMO is given by $\bar{\delta}_{k,i} = \epsilon \, \text{sgn}(\nabla \mathcal{L}_{k,i})$, where $\nabla \mathcal{L} = \nabla_\delta \mathcal{L}(x + \delta_k, y)$.

## 3. Distortion of Frank-Wolfe Attacks

We refer to the $\ell_2$ norm of an $\ell_\infty$ attack as distortion in this section we explore the insights distortion can give us into the behavior of FW-AT. For the remainder of this section FW-attacks will refer to $\ell_\infty$ attacks. In this setting, the maximal distortion possible is $\epsilon\sqrt{d}$, and we refer to $\|\delta\|_2/(\epsilon\sqrt{d})$ as the distortion ratio (or simply distortion) of the attack $\delta$.

### 3.1. Adversarial Training Rapidly Increases Distortion

Low distortion of FW-attacks was observed for standard models in (Chen et al., 2020) but robust models were not considered. We analyze the distortion ratio of FW attacks on three architectures trained with standard cross-entropy training and PGD(10)-AT on CIFAR-10. We analyze the

distortion ratio of $\epsilon = 8/255$ attacks using FW(20) on the CIFAR10 test dataset. Figure 1a shows that, while adversarial perturbations of standard models have small distortion, robust models produce attacks that are nearly maximally distorted. This phenomena occurs across three different architectures and is further supported by our theory below. We note for PGD attacks the distortion ratio can be trivially maximized with a large step size $\alpha$, and thus this connection between distortion and robustness does not exist for PGD optimization. In Figure 1b we run three epochs of FW-AT for varying number of steps and monitor the distortion. Initially there is high variation but after merely one epoch the distortion of the FW attacks has converged. As we will see in Theorem 1 this indicates that the loss gradients are rapidly aligning during FW-AT.

### 3.2. Multistep, High Distortion Attacks are Inefficient

Our main tool in analyzing the distortion of FW attacks, and a prime reason FW-AT is more mathematically transparent than PGD-AT, is a representation of the FW attack as a convex combination of the LMO iterates. We refer to the steps taken during the optimization as the attack path.

**Proposition 1.** *The FW Attack with step sizes $\gamma_k = c/(c + k)$ for some $c \geq 1$ yields the following adversarial perturbation after $K$ steps*

$$\delta_K = \epsilon \sum_{l=0}^{K-1} \alpha_l sgn(\nabla_\delta \mathcal{L}(x + \delta_l, y)) \qquad (3)$$

*where $\alpha_l = \gamma_l \prod_{i=l+1}^{K-1}(1 - \gamma_i) \in [0, 1]$ are non-decreasing in $l$, and sum to unity.*

Using this representation we can derive connections between the distortion of the attack and the variation along the attack path.

**Theorem 1.** *Consider a FW attack $\delta_K$. Let $\cos \beta_{lj}$ be the directional cosine between $sgn(\nabla_\delta \mathcal{L}(x + \delta_l, y))$ and $sgn(\nabla_\delta \mathcal{L}(x + \delta_j, y))$. The distortion ratio of the adversarial perturbation $\delta_K$ is:*

$$\frac{\|\delta_K\|_2}{\epsilon\sqrt{d}} = \sqrt{1 - 2\sum_{l<j} \alpha_l \alpha_j (1 - \cos \beta_{lj})} \qquad (4)$$

We can summarize the spirit of Theorem 1 by:

*Higher distortion is equivalent to lower gradient variation throughout the attack path.*

Concretely, the accumulation of sign changes between every step of the attack decreases distortion. Following this logic further we are able quantify the distance between different step attacks in terms of the final distortion.

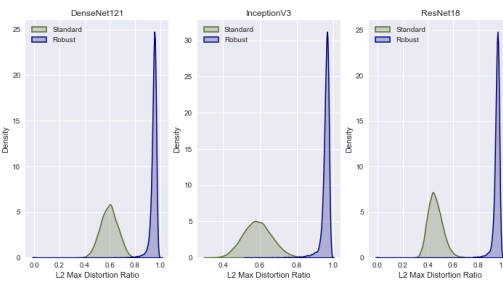

(a) Distortion ratios of FW(20) attacks

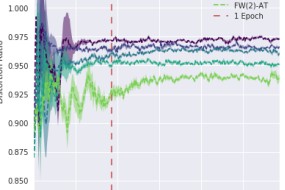

(b) Distortion during FW-AT

*Figure 1.* (1a) Cumulative density function (CDF) of $\ell_2$ distortion ratios of FW(20) attacks against standard / robust models (olive / navy) for three network architectures shows FW attacks are nearly maximally distored for robust models. (1b) Distortion of FW(k) attack against a model during its first 3 epochs of FW(k)-AT shows distortion ratios quickly converge during FW-AT.

**Theorem 2.** *Let the same conditions as Theorem 1 hold and $K > 1$. Assume the distortion ratio of the adversarial perturbation satisfies:*

$$\frac{\|\delta_K\|_2}{\epsilon\sqrt{d}} \geq \sqrt{1-\eta} \quad with \quad \eta \in (0,1).$$

*Then, the deviation of $\delta_K$ and intermediate steps $\delta_{k_0}$, for $1 \leq k_0 \leq K$ satisfy:*

$$\frac{\|\delta_K - \delta_{k_0}\|_2}{\epsilon\sqrt{d}} \leq C_{k_0,K}\sqrt{\eta} \tag{5}$$

The consequences of 2 can be summarized as:

*Multistep attacks with high distortion are inefficient.*

This suggests that during FW-AT using a large number of steps to approximate the adversarial risk results in diminishing returns once high distortion of the attacks is attained. Figure 1b shows that the distortion of attacks reaches a maximal value in the early stages of training suggesting that much of the computation used in the later steps of the optimization is wasted. Inspired by this insight we design a FW-AT algorithm which drops the number of attack steps based the distortion of a multistep attack (FW-AT-Adapt). Additional bounds are shown in the Appendix which assert that, for batches with high distortion, the gradients and thus the weight updates obtained by large step attacks are near those of a low step attack. Thus, FW-AT-Adapt is expected to achieve a similar level of robustness to non-adaptive methods. Proofs are included in the Appendix.

## 4. Frank-Wolfe Adversarial Training Algorithm

Details of Adaptive Frank-Wolfe adversarial training method (FW-AT-Adapt-E) are provided in Algorithm 2. The overall algorithm is almost identical to PGD-AT with two slight modifications:

---

**Algorithm 1** FW-Attack$(x, y; K, p)$ of size $\epsilon$

---

**Input:** Network $f_\theta$, input $(x, y)$, steps $K$.
$\delta = 0$
**for** $0 \leq k < K$ **do**
  $\bar{\bar{\delta}} = \epsilon \, \text{sgn}(\nabla_\delta \mathcal{L}(f_\theta(x + \delta), y))$
  $\delta = \delta + \frac{c}{c+k}(\bar{\bar{\delta}} - \delta)$
**end for**
**Return:** $\delta$

---

1. The adversarial attack is approximated via a FW optimization scheme (Alg. 1)

2. For the first $N_b$ batches of each epoch, the distortion of a $K$ step attack is monitored. If its mean across these batches is above a threshold $r$ then the number of attack steps is dropped to $K_0$ for the remainder of the epoch.

We also experiment with periodic distortion monitoring (FW-AT-Adapt-P) where we perform a distortion check on a single batch when the epoch is 0, 30, and 60% complete. Theorem 2 guarentees that high-step high distortion attacks are near low step attacks implying that FW-AT-Adapt should train similarly robust models to AT. A detailed expression of this can be found in Cor. 1 in the appendix.

## 5. Experimental Results

We evaluate the performance of our proposed FW-AT-Adapt against standard training, and PGD-AT (Madry et al., 2018). All networks were trained by fine-tuning for 30 epochs off a standard model via SGD optimization with a learning rate of 0.1 which was decreased to 0.01 after 15 epochs. We record the time to train the full 30 epochs and models were selected based on their checkpoint which obtained the highest adversarial accuracy on the test set. Each configuration is run 3 times and the median time is reported. The performance metric used is the accuracy on the CIFAR-10 test set after

**Algorithm 2** Epoch of FW-AT-Adapt-E with attack size $\epsilon$

---

**Input:** Network $f_\theta$, data $\mathcal{D}$, epoch learning rate $\eta_t$, high steps $K_1$, low steps $K_0$, max distortion $r$, number of monitoring batches $B_m$.

**Result:** Robust model weights $\theta$.

$N_b, d_m = 0$

$K = K_1$

**for** each batch $(x, y) \sim \mathcal{D}$ **do**

   $\delta = \text{FW-Attack}(x, y; K, p = \infty)$

   **if** $N_b \leq B_m$ **then**

      $d_m = d_m + \|\delta\|_2$

      **if** $N_b = B_m$ and $d_m / B_m > r$ **then**

         $K = K_0$

      **end if**

   **end if**

   $\theta = \theta - \eta_t \frac{1}{|B|} \sum_{i \in B} \nabla_\theta \mathcal{L}(f_\theta(x_i + \delta_i), y_i)$

   $N_b = N_b + 1$

**end for**

---

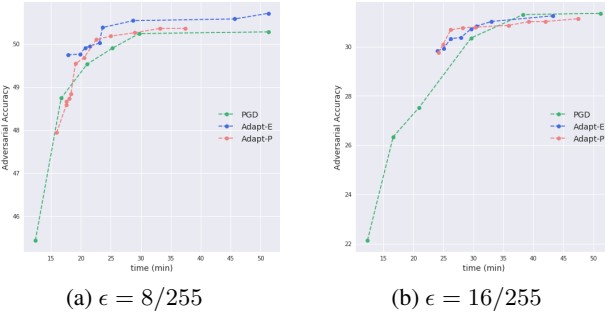

(a) $\epsilon = 8/255$      (b) $\epsilon = 16/255$

*Figure 2.* Robustness / train-time trade off performance fronts for Epoch and Periodic Adaptive FW-AT (blue / coral) shows superior trade off points compared to PGD-AT (green). Evaluations were against PGD(50) .

the attack is applied (adversarial accuracy). All PGD(K) attacks of size $\epsilon$ with are computed with step size $2.5\epsilon/K$ similar to (Madry et al., 2018).

To determine whether there are step size, drop step, and max distortion ratio combinations, $K$-$K_0$-$r$, for which FW-AT-Adapt-E/P achieves superior robustness/train time trade offs compared to PGD-AT, a hyperparameter sweep across $K \in \{4, 5, 7, 10\}$, $K_0 \in \{1, 2, 3\}$, and $r \in \{0.91, 0.92, \ldots, 0.96\}$ are compared against PGD(1)-through PGD(7)-AT, and PGD(10)-AT. We set $B_m = 3$ for all FW-AT-Adapt-E models. Based on Figure1b we allow the models to do traditional FW-AT for one epoch and then begin the adaptation. We note on CIFAR-10 AT with $\epsilon = 8/255$ performance of AT seems to saturate with a low number of steps so we experimented with adding an extra epoch of FW-AT before beginning adaptation.We then evaluated all models against a PGD(50) attack.

Figure 2 shows the performance front which start with the lowest training time models and adds models sequentially in time if their adversarial accuracy is strictly greater than the previous point on the front. Both FW-AT-Adapt-E and FW-AT-Adapt-P exhibit superior performance trade-offs to PGD-AT. Notably in the case of $\epsilon = 8/255$ (Fig. 2a ) FW-Adapt-E's performance front is strictly superior to PGD-AT's achieving points with the same robustness as PGD(10)-AT at half the training time. For $\epsilon = 16/255$ PGD(10)-AT (Fig. 2b ) remains optimal with respect to adversarial accuracy but FW-Adapt can still achieve competitive robustness at half the training time.

Gradient masking is evaluated using a similar evaluation protocol as in (Moosavi-Dezfooli et al., 2019) inspired by (Uesato et al., 2018). Table 1 shows that our method achieves high adversarial accuracy when evaluated against PGD attacks of increasing strength. Our models are evaluated against Square (Andriushchenko et al., 2020); a black-box gradient-free attack repeatedly perform queries to construct adversarial images. Table 1 contains the adversarial accuracy obtained under PGD(100) white-box attack and black-box attacks against $\ell_\infty$-robust FW-Adapt and PGD(10) models for a batch of 1000 test examples. For $\epsilon = 8/255$, FW-Adapt-E was used with parameters 10-2-0.94, and for $\epsilon = 16/255$, FW-Adapt-E parameters were 10-2-0.91. The proposed FW-AT-Adapt defense achieves similar robustness to PGD-AT. Further gradient masking evaluations are included in the Appendix.

| Method | $\epsilon$ | Clean | PGD(100) | Square | Time (mins) |
|---|---|---|---|---|---|
| FW-AT-Adapt-E | 8/255 | 82.53 | 50.70 | 54.60 | 33.06 |
| PGD-AT | 8/255 | 82.91 | 49.70 | 55.10 | 51.85 |
| FW-AT-Adapt-E | 16/255 | 62.65 | 32.40 | 34.40 | 36.09 |
| PGD-AT | 16/255 | 62.14 | 31.70 | 34.20 | 52.70 |

*Table 1.* Model accuracy on CIFAR-10 test set for white-box $\ell_\infty$ PGD(100) attacks and black-box attacks (Square) at $\epsilon = 8/255, 16/255$ for $\ell_\infty$ robust networks trained on ResNet18 architecture at $\epsilon = 8/255, 16/255$ respectively.

## 6. Conclusion

FW $\ell_\infty$ attacks against robust models have higher $\ell_2$ distortions than standard ones. We derive a theoretical connection between loss geometry and distortion of FW attacks which explains this phenomenon and bound the variation along the attack path in terms of the distortion. Inspired by this connection, we propose an adaptive AT algorithm (FW-AT-Adapt) that achieves high robustness with lower training time than PGD-AT against strong white- and black-box attacks, and is resistant to gradient obfuscation. We hope this work encourages future research on the connection between Frank-Wolfe optimization and adversarial robustness.

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

# 7. Appendix

## 7.1. Theoretical Results and Proofs

While loss functions $\mathcal{L}(x+\delta, y)$ in deep neural networks are non-convex in general, we make the following assumption.

**Assumption 1.** *The function $\mathcal{L}$ has $L$-Lipschitz continuous gradients on $B_p(\epsilon)$, i.e., $\|\nabla\mathcal{L}(x+u, y) - \nabla\mathcal{L}(x+v, y)\| \leq L\|u - v\|, \forall u, v \in B_p(\epsilon)$.*

Assumption 1 is a standard assumption for the non-convex setting and has been made in several works (Lacoste-Julien, 2016; Chen et al., 2020). For standard ERM training loss smoothing has been observed to occur in over parameterized (Allen-Zhu et al., 2019; Zou & Gu, 2019; Cao & Gu, 2020) and batch normalized (Santurkar et al., 2018) DNNs. For robust models the process of adversarial training has been noted to significantly smooth the loss (Moosavi-Dezfooli et al., 2019; Qin et al., 2019). an example comparing standard and robust loss landscapes is given in Figure 3. Moreover, the distortion plots of Figure 1 (b) together with Theorem 1 suggest that FW-AT quickly regularizes the loss landscape.

**Corollary 1.** *Consider a batch update of FW-AT Algorithm 2 where the high distortion condition of Thm. 2 holds on average on examples in a batch $B$, i.e. for some small $\eta \in (0,1)$:*

$$\frac{1}{|B|} \sum_{i \in B} \frac{\|\delta_i(\theta, K)\|_2}{\epsilon\sqrt{d}} \geq \sqrt{1 - \eta} \qquad (6)$$

*where $\delta_i(\theta, K)$ denotes the $K$-step FW adversarial perturbation for the $i$-th example in the batch $B$. Let the SGD model weight gradient be given by:*

$$g(\theta, \delta(\theta, K)) = \frac{1}{|B|} \sum_{i \in B} \nabla_\theta \mathcal{L}(f_\theta(x_i + \delta_i(\theta, K)), y_i)$$

*Given Assumption 1 holds, the model weights SGD update using adversarial perturbations $\delta_K$ and $\delta_{k_0}$ are bounded as:*

$$\|g(\theta, \delta(\theta, K)) - g(\theta, \delta(\theta, k_0))\|_2 \leq LC_{k_0, K}\sqrt{\eta} \cdot \epsilon\sqrt{d}. \quad (7)$$

*where $L$ is the local Lipschitz constant.*

Bound (7) asserts that in the high distoriton setting the gradients, and thus the weight updates, obtained by a high step attack are near those of a low step attack. Therfore, it is expected to achieve a similar level of adversarial robustness using the proposed adaptive algorithm.

## 7.2. Loss Landscape

It has been shown experimentally that AT robust models and geometric regularization methods that increase adversarial robustness, have more regular loss landscapes than

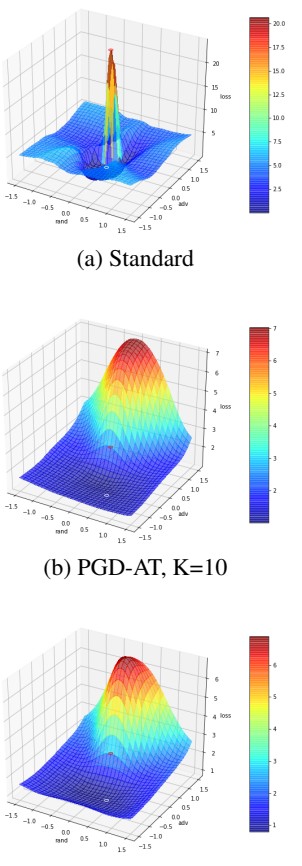

(a) Standard

(b) PGD-AT, K=10

(c) FW-AT-Adapt-E, 10-2-0.94

*Figure 3.* Loss landscapes for an image from CIFAR10 test set. ResNet18 architecture. Standard models have highly non-linear loss surfaces as opposed to robust models which have smoother landscapes.

their non-robust counterparts (Lyu et al., 2015; Ros & Doshi-Velez, 2018; Moosavi-Dezfooli et al., 2019; Qin et al., 2019). Figure 3 shows sample loss surfaces for a standard model, PGD(10)-AT, and FW(10)-AT. The white circle denotes the original image, and the red circle denotes the $\ell_\infty$ adversarial perturbation at $\epsilon = 8/255$. One axis is aligned to the adversarial perturbation direction and the other is aligned to a random orthogonal direction. The loss surface of standard models is highly non-linear and the loss varies significantly in this small neighborhood, with the largest variation occurs in the adversarial direction. Figure 3 (b) and (c) demonstrate that robust models offer more resistance to such local loss variation and the variation is primarily in the direction of the adversarial attack. These results suggest high resistance to gradient obfuscation.

## 7.3. Adversarial Robustness: White-Box Evaluations

To further evaluate FW-AT-Adapt we choose strong performers from the previous experiment to test against a

larger variety of adversarial attacks, including white-box untargeted and targeted attacks towards a random class $r \sim \mathcal{U}(\{1, \ldots, K\} \backslash y)$. The white-box setup considers attack techniques that have full access to the model parameters and are constrained by the same maximum perturbation size $\epsilon$. The classification margin is defined as $M(x, y) = \log p_y(x) - \max_{j \neq y} \log p_j(x)$. The following white-box attacks are used for evaluating adversarial robustness:

**(UL) Untargeted-loss:** $\max_{\delta \in B(\epsilon)} \mathcal{L}(x + \delta, y)$
**(TL) Random Targeted-loss:** $\min_{\delta \in B(\epsilon)} \mathcal{L}(x + \delta, r)$
**(UM) Untargeted-margin:** $\min_{\delta \in B(\epsilon)} M(x + \delta, y)$
**(TM) Random Targeted-margin:** $\max_{\delta \in B(\epsilon)} M(x + \delta, r)$

Table 2 reports robustness results on CIFAR-10 for these $\ell_\infty$ attacks at $\epsilon = 8/255, 16/255$ for a ResNet18 architecture trained at $\epsilon = 8/255, 16/255$ respectively. FW-AT-Adapt achieves robustness competitive with PGD-AT for untargeted and targeted attacks at lower training times.

### 7.4. Gradient Masking Evaluations

Figure 4 compares the margin computed using Square attack and PGD(100) attack for a large batch of test points. The margin captures confidence, and is positive for correct predictions and negative for misclassifications. The margin was computed with black-box Square Attack for the y-axis and white-box PGD(100) attack for x-axis on a set of 1000 test points. Points near the line $y = x$ indicate both types of attacks found similar adversarial perturbations, while points below the line shown in red imply that Square Attack identified stronger attacks than PGD. It is observed that both methods lead to a similar margin except on a small subset of points for FW-AT-Adapt-E. For $\epsilon = 8/255$, FW-AT-Adapt-E has $13/1000 = 1.3\%$ and PGD-AT has $23/1000 = 2.3\%$ points in red. For $\epsilon = 16/255$, FW-AT-Adapt-E has $130/1000 = 13\%$ and PGD-AT has $120/1000 = 12\%$ points in red. The results in Table 1 and Fig. 4 further verify that FW-AT improves the true robustness and does not suffer from grading masking or obfuscation.

### 7.5. Proof of Proposition 1

*Proof.* The LMO solution is given by $\bar{\delta}_k = \epsilon \, \phi_p(\nabla_\delta \mathcal{L}(x + \delta_k, y))$ and the update becomes

$$\delta_{k+1} = \delta_k + \gamma_k(\bar{\delta}_k - \delta_k)$$
$$= (1 - \gamma_k)\delta_k + \gamma_k \, \epsilon \, \phi_p(\nabla_\delta \mathcal{L}(x + \delta_k, y))$$

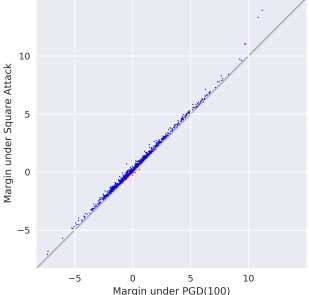

(a) PGD-AT, K=10, $\epsilon = 8/255$

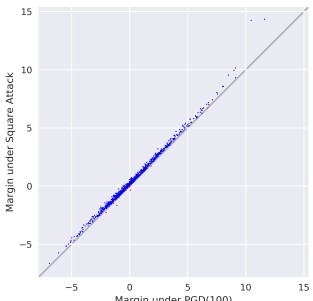

(b) FW-AT-Adapt-E, 10-2-0.94, $\epsilon = 8/255$

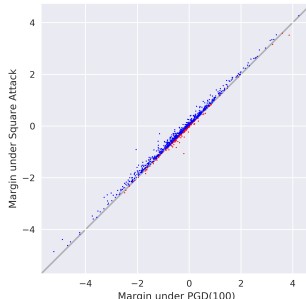

(c) PGD-AT, K=10, $\epsilon = 16/255$

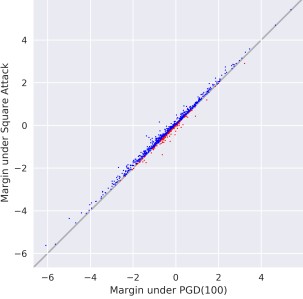

(d) FW-AT-Adapt-E, 10-2-0.91, $\epsilon = 16/255$

*Figure 4.* Gradient masking analysis for CIFAR10 PGD-AT and FW-AT-Adapt-E $\ell_\infty$ robust networks trained on ResNet18 architecture. FW-AT-Adapt-E exhibits a high resistance to gradient masking.

| Method ($\epsilon = 8/255$) | Clean | UL | TL | UM | TM | Params | Time (mins) |
|---|---|---|---|---|---|---|---|
| Standard | 94.35 | 0.00 | 10.90 | 0.00 | 12.07 | n/a | n/a |
| FW-AT-Adapt-P | 84.07 | 49.85 | 75.84 | 48.61 | 73.84 | 5-2-0.91 | 19.13 |
| FW-AT-Adapt-E | 83.11 | 50.44 | 75.63 | 48.78 | 73.16 | 10-3-0.92 | 23.57 |
| FW-AT-Adapt-E | 82.53 | 50.66 | 74.80 | 48.39 | 72.93 | 10-2-0.94 | 33.06 |
| PGD-AT | 82.06 | 49.66 | 74.49 | 48.23 | 72.98 | 3 | 21.04 |
| PGD-AT | 82.60 | 50.35 | 75.66 | 48.84 | 73.27 | 5 | 29.71 |
| PGD-AT | 82.91 | 50.41 | 75.21 | 48.86 | 73.62 | 10 | 51.85 |
| Method ($\epsilon = 16/255$) | Clean | UL | TL | UM | TM | Params | Time (mins) |
| FW-AT-Adapt-P | 62.21 | 31.14 | 54.30 | 27.33 | 52.30 | 4-2-0.92 | 24.94 |
| FW-AT-Adapt-E | 60.85 | 30.93 | 53.49 | 26.63 | 51.23 | 7-2-0.91 | 30.48 |
| FW-AT-Adapt-E | 62.65 | 31.10 | 54.41 | 27.29 | 52.40 | 10-2-0.91 | 36.09 |
| PGD-AT | 57.18 | 27.60 | 49.81 | 22.83 | 47.81 | 3 | 20.97 |
| PGD-AT | 61.81 | 30.46 | 53.76 | 27.21 | 52.38 | 5 | 29.64 |
| PGD-AT | 61.83 | 31.48 | 54.30 | 27.34 | 52.09 | 10 | 52.70 |

*Table 2.* Model accuracy on CIFAR-10 test set against $\ell_\infty$ attacks at $\epsilon = 8/255, 16/255$ on ResNet18 architecture trained at $\epsilon = 8/255, 16/255$ respectively against PGD(50) attacks step size $\alpha = 2.5\epsilon/50$. Our proposed adaptive FW-AT-Adapt algorithm obtains high robustness similar to PGD(10)-AT at lower training time.

Using induction on this relation yields after $K$ steps:

$$\delta_K = \delta_0 \prod_{l=0}^{K-1}(1 - \gamma_l)$$

$$+ \epsilon \sum_{l=0}^{K-1} \gamma_l \prod_{i=l+1}^{K-1}(1 - \gamma_i)\phi_p(\nabla_\delta \mathcal{L}(x + \delta_k, y)) \quad (8)$$

where $\delta_0$ is the initial point which affects both terms in (8) and $\gamma_k = c/(c + k)$ for $k \geq 0$. Since $\gamma_0 = 1$, the first term vanishes and (8) simplifies to

$$\delta_K = \epsilon \sum_{l=0}^{K-1} \alpha_l \phi_p(\nabla_\delta \mathcal{L}(x + \delta_l, y)) \quad (9)$$

where the coefficients are

$$\alpha_l = \gamma_l \prod_{i=l+1}^{K-1}(1 - \gamma_i) \quad (10)$$

Since $\gamma_l \in [0, 1]$, it follows that $\alpha_l \in [0, 1]$. Induction on (10) yields that $\sum_{l=0}^{K-1} \alpha_l = 1$. Furthermore, $\alpha_l \leq \alpha_{l+1}$ follows from:

$$\alpha_l \leq \alpha_{l+1}$$
$$\Leftrightarrow \gamma_l(1 - \gamma_{l+1}) \leq \gamma_{l+1}$$
$$\Leftrightarrow \frac{c}{c+l}\left(1 - \frac{c}{c+l+1}\right) \leq \frac{c}{c+l+1}$$
$$\Leftrightarrow \frac{l+1}{c+l} \leq 1$$
$$\Leftrightarrow 1 \leq c$$

Thus, the sequence $\alpha_l$ is non-decreasing in $l$. Since the coefficients sum to unity, (9) is in the convex hull of the generated LMO sequence $\{\phi_p(\nabla_\delta \mathcal{L}(x + \delta_l)) : l = 0, \ldots, K - 1\}$. □

### 7.6. Proof of Theorem 1

*Proof.* From Proposition 1, we obtain the following decomposition of the adversarial perturbation:

$$\delta_K = \epsilon \sum_{l=0}^{K-1} \alpha_l \mathrm{sgn}(\nabla_\delta \mathcal{L}(x + \delta_l, y))$$

To bound the magnitude of the adversarial perturbation, we have

$$\|\delta_K\|_2 = \sqrt{\|\delta_K\|_2^2} = \epsilon \sqrt{\left\|\sum_l \alpha_l s_l\right\|_2^2}$$

where we use the shorthand notation $s_l = \mathrm{sgn}(\nabla_\delta \mathcal{L}(x + \delta_l, y))$. The squared $\ell_2$ norm in the above is bounded as:

$$\left\|\sum_l \alpha_l s_l\right\|_2^2 = \sum_l \sum_j \alpha_l \alpha_j \langle s_l, s_j \rangle$$
$$= \sum_l (\alpha_l)^2 \|s_l\|_2^2 + \sum_{l \neq j} \alpha_l \alpha_j \|s_l\|_2 \|s_j\|_2 \cos \beta_{lj}$$
$$= d\left(\sum_l (\alpha_l)^2 + \sum_{l \neq j} \alpha_l \alpha_j \cos \beta_{lj}\right)$$
$$= d\left(\sum_l (\alpha_l)^2 + \sum_{l \neq j} \alpha_l \alpha_j - \sum_{l \neq j} \alpha_l \alpha_j (1 - \cos \beta_{lj})\right)$$
$$= d\left(1 - \sum_{l \neq j} \alpha_l \alpha_j (1 - \cos \beta_{lj})\right)$$
$$= d\left(1 - 2\sum_{l < j} \alpha_l \alpha_j (1 - \cos \beta_{lj})\right)$$

where we used $\|s_l\|_2 = \sqrt{d}$ and from Proposition 1 $(\sum_l \alpha_l)^2 = 1$. The final step follows from symmetry. This concludes the proof. □

## 7.7. Proof of Theorem 2

*Proof.* From Theorem 1 and the lower bound on the distortion, it follows that:

$$\sum_{l<j} \alpha_l \alpha_j (1 - \cos \beta_{lj}) \leq \eta/2 \tag{11}$$

Letting $s_i = \text{sgn}(\nabla \mathcal{L}(x + \delta_i, y))$ and expanding the squared difference of signed gradients:

$$\begin{aligned}
\|s_l - s_j\|_2^2 &= \|s_l\|_2^2 + \|s_j\|_2^2 - 2\langle s_j, s_l \rangle \\
&= \|s_l\|_2^2 + \|s_j\|_2^2 - 2\|s_j\|_2 \|s_l\|_2 \cos \beta_{lj} \\
&= d + d - 2d \cos \beta_{lj} \\
&= 2d(1 - \cos \beta_{lj})
\end{aligned} \tag{12}$$

Using (12) into (11),

$$\sum_{l<j} \alpha_l \alpha_j \|s_l - s_j\|_2^2 \leq \eta d \tag{13}$$

For the FGSM deviation bound, i.e., $k_0 = 1$, we have by the triangle inequality:

$$\begin{aligned}
\|\delta_K - \epsilon \text{sgn}(\nabla_x \mathcal{L}(x,y))\|_2 &= \|\epsilon \sum_{l=0}^{K-1} \alpha_l s_l - \epsilon s_0\|_2 \\
&= \|\epsilon \sum_l \alpha_l s_l - \sum_l \alpha_l \epsilon s_0\|_2 \\
&= \epsilon \|\sum_l \alpha_l (s_l - s_0)\|_2 \\
&\leq \epsilon \sum_{l>0} \alpha_l \|s_l - s_0\|_2
\end{aligned} \tag{14}$$

Using Cauchy-Schwarz inequality, we obtain:

$$\begin{aligned}
\sum_{l>0} \alpha_l \|s_l - s_0\|_2 &\leq \sqrt{K-1} \sqrt{\sum_{l>0} (\alpha_l)^2 \|s_l - s_0\|_2^2} \\
&\leq \sqrt{K-1} \sqrt{\sum_{l<j} (\alpha_l)^2 \|s_l - s_j\|_2^2} \\
&\leq \sqrt{K-1} \sqrt{\sum_{l<j} \alpha_l \alpha_j \|s_l - s_j\|_2^2} \\
&\leq \sqrt{K-1} \cdot \sqrt{\eta d}
\end{aligned} \tag{15}$$

where we used the non-decreasing property of the sequence $\{\alpha_l\}_l$ and the bound (13). This concludes the first part.

Given $1 \leq k_0 \leq K$, we have via using Proposition 1 twice:

$$\begin{aligned}
\delta_K - \delta_{k_0} &= \epsilon \sum_{l=0}^{K-1} \alpha_l s_l - \delta_{k_0} \\
&= \epsilon \sum_{l=0}^{K-1} \alpha_l s_l - \sum_l \alpha_l \delta_{k_0} \\
&= \epsilon \sum_{l=0}^{K-1} \alpha_l (s_l - \delta_{k_0}/\epsilon) \\
&= \epsilon \sum_{l=0}^{K-1} \alpha_l (s_l - \sum_{j=0}^{k_0-1} \tilde{\alpha}_j s_j) \\
&= \epsilon \sum_{l=0}^{K-1} \alpha_l \sum_{j=0}^{k_0-1} \tilde{\alpha}_j (s_l - s_j) \\
&= \epsilon \sum_{l=0}^{K-1} \sum_{j=0}^{k_0-1} \alpha_l \tilde{\alpha}_j (s_l - s_j)
\end{aligned} \tag{16}$$

where $\alpha_l = \gamma_l \prod_{i=l+1}^{K-1} (1 - \gamma_i), 0 \leq l \leq K-1$ and $\tilde{\alpha}_j = \gamma_j \prod_{i=l+1}^{k_0-1} (1 - \gamma_i), 0 \leq j \leq k_0 - 1$.

Taking the $\ell_2$ norm of both sides of (16) and using the triangle inequality, we obtain:

$$\|\delta_K - \delta_{k_0}\|_2 \leq \epsilon \sum_{l=0}^{K-1} \sum_{j=0}^{k_0-1} \alpha_l \tilde{\alpha}_j \|s_l - s_j\|_2$$

Using the Cauchy-Schwarz inequality yields:

$$\begin{aligned}
&\sum_{l=0}^{K-1} \sum_{j=0}^{k_0-1} \alpha_l \tilde{\alpha}_j \|s_l - s_j\|_2 \\
&\leq \sqrt{\sum_{l=0}^{K-1} \sum_{j=0}^{k_0-1} (\alpha_l \tilde{\alpha}_j)^2} \sqrt{\sum_{l=0}^{K-1} \sum_{j=0}^{k_0-1} \|s_l - s_j\|_2^2} \\
&\leq \sqrt{\sum_{l=0}^{K-1} (\alpha_l)^2 \sum_{j=0}^{k_0-1} (\tilde{\alpha}_j)^2} \sqrt{\sum_{l=0}^{K-1} \sum_{j=0}^{K-1} \|s_l - s_j\|_2^2} \\
&\leq \sqrt{\sum_{l=0}^{K-1} (\alpha_l)^2 \sum_{j=0}^{k_0-1} (\tilde{\alpha}_j)^2} \sqrt{\frac{2\eta d}{\min_{l<j}\{\alpha_l \alpha_j\}}} \\
&= \sqrt{\frac{2 \sum_{l=0}^{K-1} (\alpha_l)^2 \sum_{j=0}^{k_0-1} (\tilde{\alpha}_j)^2}{\alpha_0 \alpha_1}} \sqrt{\eta d}
\end{aligned}$$

where we used (13) and the non-decreasing sequence $\{\alpha_l\}$ implies $\min_{l<j}\{\alpha_l \alpha_j\} = \alpha_0 \alpha_1$. This concludes the proof of the second part. $\square$

### 7.8. Proof of Corollary 1

*Proof.* Using the triangle inequality and the $L$-Lipschitz continuous loss gradient assumption:

$$\|g(\theta, \delta(\theta, K)) - g(\theta, \delta(\theta, k_0))\|_2$$
$$= \|\frac{1}{|B|} \sum_{i \in B} (\nabla_\theta \mathcal{L}(f_\theta(x_i + \delta_i(\theta, K)), y_i)$$
$$- \nabla_\theta \mathcal{L}(f_\theta(x_i + \delta_i(\theta, k_0)), y_i))\|_2$$
$$\leq \frac{1}{|B|} \sum_{i \in B} \|\nabla_\theta \mathcal{L}(f_\theta(x_i + \delta_i(\theta, K)), y_i)$$
$$- \nabla_\theta \mathcal{L}(f_\theta(x_i + \delta_i(\theta, k_0)), y_i))\|_2$$
$$\leq \frac{L}{|B|} \sum_{i \in B} \|\delta_i(\theta, K) - \delta_i(\theta, k_0)\|_2 \qquad (17)$$

The average distortion condition yields via Proposition 1 (with the superscript $(i)$ denoting the $i$-th example variables):

$$\frac{1}{|B|} \sum_{i \in B} \sqrt{1 - 2 \sum_{l < j} \alpha_l \alpha_j (1 - \cos \beta_{lj}^{(i)})} \geq \sqrt{1 - \eta}$$

Using Jensen's inequality (and the concavity of the square root function) further yields after some algebra:

$$\frac{1}{|B|} \sum_{i \in B} \sum_{l < j} \alpha_l \alpha_j (1 - \cos \beta_{lj}^{(i)}) \leq \frac{\eta}{2}$$

Borrowing the relation (12) from the proof of Theorem 2, we further obtain:

$$\frac{1}{|B|} \sum_{i \in B} \sum_{l < j} \alpha_l \alpha_j \|s_l^{(i)} - s_j^{(i)}\|_2^2 \leq \eta d \qquad (18)$$

Using the relation (16), it follows:

$$\frac{1}{|B|} \sum_{i \in B} \|\delta_i(\theta, K) - \delta_i(\theta, k_0)\|_2$$
$$\overset{(a)}{\leq} \frac{1}{|B|} \sum_{i \in B} \epsilon \sum_{l=0}^{K-1} \sum_{j=0}^{k_0-1} \alpha_l \tilde{\alpha}_j \|s_l^{(i)} - s_j^{(i)}\|_2$$
$$\overset{(b)}{\leq} \epsilon \sqrt{\sum_{l=0}^{K-1} \alpha_l^2 \sum_{j=0}^{k_0-1} \tilde{\alpha}_j^2} \cdot \frac{1}{|B|} \sum_{i \in B} \sqrt{\sum_{l=0}^{K-1} \sum_{j=0}^{k_0-1} \|s_l^{(i)} - s_j^{(i)}\|_2}$$
$$\overset{(c)}{\leq} \epsilon \sqrt{\sum_{l=0}^{K-1} \alpha_l^2 \sum_{j=0}^{k_0-1} \tilde{\alpha}_j^2} \cdot \sqrt{\frac{1}{|B|} \sum_{i \in B} \sum_{l=0}^{K-1} \sum_{j=0}^{k_0-1} \|s_l^{(i)} - s_j^{(i)}\|_2}$$
$$\qquad (19)$$

where we used (a) triangle inequality, (b) Cauchy-Schwarz and (c) Jensen's inequality.

From (18), it follows that:

$$\frac{1}{|B|} \sum_{i \in B} \sum_{l=0}^{K-1} \sum_{j=0}^{k_0-1} \|s_l^{(i)} - s_j^{(i)}\|_2^2 \leq \frac{2\eta d}{\alpha_0 \alpha_1} \qquad (20)$$

Combining (20) with (19) yields:

$$\frac{1}{|B|} \sum_{i \in B} \|\delta_i(\theta, K) - \delta_i(\theta, k_0)\|_2 \leq \epsilon \sqrt{d} \sqrt{\eta} C_{k_0, K}$$

where $C_{k_0, K} = \sqrt{\frac{2 \sum_{l=0}^{K-1} \alpha_l^2 \sum_{j=0}^{k_0-1} \tilde{\alpha}_j^2}{\alpha_0 \alpha_1}}$. Using this bound in (17) concludes the proof. $\square$

### 7.9. Convergence Analysis

Loss functions $\mathcal{L}(x + \delta, y)$ in deep neural networks are non-convex in general. For a targeted attack that aims to fool the classifier to predict a specific label, without loss of generality, we seek to minimize the loss $f(\delta) = \mathcal{L}(x + \delta, y')$ over a $\ell_p$ constraint set. The untargeted case follows similarly. [1] For general non-convex constrained optimization, the Frank-Wolfe gap given by (Frank & Wolfe, 1956):

$$G(\delta_k) = \max_{\delta \in B_p(\epsilon)} \langle \delta - \delta_k, \nabla_\delta \mathcal{L}(x + \delta_k, y) \rangle \qquad (21)$$

is non-negative in general and zero at stationary points. The convergence of FW on non-convex functions has been studied in (Lacoste-Julien, 2016) and recently improved for strongly convex constraints in (Rector-Brooks et al., 2019).

**Assumption 2.** *The function $f$ has $L$-Lipschitz continuous gradients on $B_p(\epsilon)$, i.e., $\|\nabla f(u) - \nabla f(v)\| \leq L\|u - v\|, \forall u, v \in B_p(\epsilon)$.*

Assumption 2 is a standard assumption for the non-convex setting and has been made in several works (Lacoste-Julien, 2016; Chen et al., 2020). A recent study (Santurkar et al., 2018) shows that the batch normalization layer used in modern neural networks makes the loss much smoother. Other recent works (Allen-Zhu et al., 2019; Zou & Gu, 2019; Cao & Gu, 2020) showed that the loss is semi-smooth for overparameterized DNNs. Furthermore, the process of adversarial training smooths the loss landscape in comparison to standard models significantly as Fig. 3 illustrates and other works have noted this phenomenon as well (Moosavi-Dezfooli et al., 2019; Qin et al., 2019).

Given Assumption 2 and the compactness of the constraint sets, all limit points of FW are stationary points (Bertsekas, 1999). The convergence rate of FW to a stationary point for optimization over arbitrary convex sets was first shown in (Lacoste-Julien, 2016) given by

$$\min_{1 \leq s \leq t} G(\delta_s) \leq \frac{\max\{2h_0, L \operatorname{diam}(B)\}}{\sqrt{t+1}}$$

where $h_0 = f(\delta_0) - \min_{\delta \in B(\epsilon)} f(\delta)$ is the initial global suboptimality. It follows that larger $\epsilon$ imply a larger diameter

---

[1] For untargeted attacks, $\min_{\delta \in B(\epsilon)} -\mathcal{L}(x + \delta, y)$ is considered and the FW gap becomes (21).

and more iterations may be needed to converge [2]. This result implies that an approximate stationary point can be found with gap less than $\epsilon_0$ in at most $O(1/\epsilon_0^2)$ iterations. Theorem 4 in (Rector-Brooks et al., 2019) shows that for smooth non-convex functions over strongly convex constraint sets, FW yields an improved convergence rate $O\left(\frac{1}{t}\right)$, which importantly does not hold for the $\ell_\infty$ constraint.

## Acknowledgements

Research was sponsored by the United States Air Force Research Laboratory and the United States Air Force Artificial Intelligence Accelerator and was accomplished under Cooperative Agreement Number FA8750-19-2-1000. The views and conclusions contained in this document are those of the authors and should not be interpreted as representing the official policies, either expressed or implied, of the United States Air Force or the U.S. Government. The U.S. Government is authorized to reproduce and distribute reprints for Government purposes notwithstanding any copyright notation herein.

---

[2]The diameter of $\ell_2$ ball is $2\epsilon$ and for the $\ell_\infty$ ball $2\epsilon\sqrt{d}$.