# OpenReview forum: "On Frank-Wolfe Adversarial Training"
_ICML.cc/2021/Workshop/AML — ICML 2021 Workshop AML Poster_

### Official Review · Reviewer_bQqH · 2021-06-20
**A promising alternative to PGD-AT**

**Rating:** Accept
**Confidence:** 3

**Review:**

The paper uses the Frank-Wolfe optimization to derive a relationship between loss landscape and attack distortion. Based on this relationship the authors propose an adaptive Frank-Wolfe adversarial training (FW-AT-Adapt) method to obtain robustness. This method can achieve robustness near or above multistep PGD-AT while significantly decreasing training time. Briefly, the paper provides a promising alternative to PGD-AT.

This work may shed light on future research on the connecting Frank-Wolfe optimization and adversarial robustness.

---

### Decision · Program_Chairs · 2021-06-21

**Decision:**

Accept (Poster)

**Comment:**

This paper proposed an adaptive Frank-Wolfe adversarial training method to improve robustness, achieving comparable robustness with PGD-AT while significantly decreasing training time.